# Traumatised Children’s Perspectives on Their Lived Experience: A Review

**DOI:** 10.3390/bs13020170

**Published:** 2023-02-14

**Authors:** Wai Tong Chien, Chi Tung Lau

**Affiliations:** The Nethersole School of Nursing, Faculty of Medicine, The Chinese University of Hong Kong, Shatin, New Territories, Hong Kong SAR, China

**Keywords:** traumatised children, trauma, lived experience, post-traumatic stress, systematic review

## Abstract

*Introduction:* Most children have exposure of traumatic events during their life, such as natural disasters, accidents, and abuses. A review of traumatised children’s perspective on traumatic events plays an important role in enhancing our understanding and promoting appropriate tailor-made intervention and support to these children. *Methods:* Four main health-related electronic databases were searched for all English full-text qualitative research articles over the past 11 years to uncover the recent best available perspective/evidence from traumatised children. The PRISMA checklist was adopted to guide the review process. *Results:* Five themes about children’s experiences and perspectives towards the traumatic events encountered were summarised and integrated from 19 qualitative studies identified. They included daily life problems related to trauma, negative responses to trauma, perceived health needs, coping strategies related to trauma and stress, and growth from traumatic experience. *Conclusions:* This systematic review provides evidence about responses/impacts and perceived health needs of traumatised children and informs the direction caregivers’ training can take, helping these children by early identification and timely intervention. More research is needed to examine/compare traumatised children’s responses and coping between diverse traumatic experiences, time from exposure, and the sociodemographic characteristics of these children.

## 1. Introduction

Traumas are long-lasting emotional responses that often result from living through a distressing event, which may subsequently contribute to post-traumatic stress disorder (PTSD) [1]. The majority of children have been exposed to different types of traumas, such as natural disasters, accidents, and abuse during their early life stages [2,3,4]. These trauma events are perceived by individual children to be frightening, violent and/or dangerous experiences, causing intense behavioural and physical reactions that likely last over a long period of time [5]. Recent studies have shown that trauma exposure is associated with elevated risks of psychiatric disorders, suicidality, substance use, psychological adjustment difficulties (e.g., PTSD), and/or functional impairments (e.g., poorer adaptive functioning and social functioning) [6,7]. Epidemiological studies and territory-wide service evaluations suggest that traumatic events can adversely affect children’s social, emotional, cognitive, and physical development, as well as their well-being [8,9,10]. These studies have contributed to our recognition of the severe and long-term impacts of traumatic events on children’s health in adulthood.

In the past, children’s views have been undervalued in medical care, as they were seen as being protected, vulnerable, and difficult to be engaged with in research [11]. Understanding children’s perceptions of their own traumas is essential to help them adjust to their traumatic experiences [12]. Over the past decade, there have been qualitative studies conducted to explore or attempt to support and guide the development of an appropriate and effective intervention addressing the complex health needs of children exposed to traumas. Although a review on the psychosocial impacts of traumatic experiences among children was published a decade ago [13], many selected articles included only parents’ or caregivers’ perspectives concerning the experiences of their traumatised children. Without children’s perspective in the qualitative evidence, it is difficult to accurately examine children’s own specific and important perspectives on their traumatic experiences and related responses and impacts. Therefore, this systematic review identified the best available qualitative research articles published over the past 11 years (between 2012 and 2022) and aimed to summarise and consolidate the current evidence from all accessible qualitative research on children’s subjective experiences and perspectives on their traumas and related daily life problems, positive and negative responses, and coping strategies used.

## 2. Materials & Methods

### 2.1. Search Strategy

Qualitative studies on children’s perspectives on traumatic experiences were searched using four main electronic databases, CINAHL, ProQuest, PubMed, and Web of Science. The keywords for the literature search were focused on trauma (e.g., trauma, traumatic, or post-traumatic stress), children (e.g., children, adolescents, or adolescence) qualitative research (e.g., qualitative, exploratory, or semi-structured/unstructured interview), and stress (e.g., stress, psychological response/reaction impact, or coping). The search covered peer-reviewed and full-text English articles published between 1 January 2012 and 30 June 2022 (i.e., to uncover the recent best available perspective/evidence from the traumatised children). The article search and selection procedure (Figure 1) was performed according to the PRISMA checklist (http://www.prisma-statement.org/; accessed on 13 July 2022).

### 2.2. Inclusion Criteria

This review included qualitative studies. The studies were included if they focused on the perspectives of traumatised children (aged ≤18 years) about their traumatic events and related experiences and responses. Within articles describing interviews of both traumatised children and their parents/caregivers/service providers, mainly the children’s views about their traumatic experiences and related stress were included.

### 2.3. Exclusion Criteria

Articles were excluded if: (a) They were adult subjects’ recollections of their long-ago childhood trauma; (b) Only involved the views of traumatised children’s parents/caregivers on their children’s traumatic experiences; (c) They did not focus upon the main objective on traumatic experiences and/or their related stress/impacts; (d) They were not based upon a qualitative research design (e.g., cross-section or longitudinal survey study, review or discussion papers, case studies, and editorials); and (e) Presented ongoing research or a brief report.

### 2.4. Study Selection Procedures

All studies identified from database searches were exported into Convidence, and duplicates were removed. The titles and abstracts of all the articles were read and initially screened for eligibility according to the above-mentioned criteria. The full texts of the potentially eligible articles were retrieved and screened by two reviewers (CTL and WTC) independently. Finally, the screening results and disagreements were discussed among the two reviewers and a third independent reviewer until all reviewers reached consensus about the results.

### 2.5. Quality Appraisal

The quality of reviewed qualitative studies was then critically examined using the critical appraisal skills programme (CASP) checklist for qualitative research [14]. Quality assessment was undertaken by two independent reviewers (CTL and WTC). Any discrepancy was discussed and an agreed decision was reached between reviewers. Quality appraisal was not used to determine excluded studies, but rather to assess each study’s quality. The overall studies were judged to be generally of good quality (see Appendix A).

### 2.6. Data Extraction and Synthesis

The study information of all included studies (e.g., authors, country, and year of publication; study design, types of traumas, main characteristics, and size of study sample) were extracted and summarised. The qualitative findings were coded, condensed, and integrated into themes and subthemes on an Excel spreadsheet. Commonalities and differences in subthemes of subjective traumatic experience in the included studies were discussed.

## 3. Results

### 3.1. Selection of Qualitative Studies

A total of 1107 studies were identified from the electronic searches on the four databases. After removing the duplicates, 844 articles remained. Of these, 814 did not meet the inclusion criteria due to adult population (*n* = 309), animal samples (*n* = 2), age > 18 youth participants (*n* = 10), not focusing on trauma experiences and its impact (*n* = 279), and non-qualitative study design (*n* = 214). The full texts of four studies were not available. After reviewing the full text of the remaining 26 relevant studies, another seven studies were excluded for several reasons, including majority of participants were adolescents, studies conducted in the adult population, and studies employing the wrong study design. Finally, the 19 articles that met all the review criteria were included. The PRISMA flow diagram of the article searching and selection process of this systematic review is presented in Figure 1.

### 3.2. Characteristics of Included Studies

The 19 included studies were conducted worldwide, including Australia (*n* = 1), Cambodia (*n* = 1), the Democratic Republic of Congo (*n* = 1), Denmark (*n* = 1), Indonesia (*n* = 1), Iran (*n* = 1), Myanmar (*n* = 1), Norway (*n* = 1), Palestine (*n* = 1), Sweden (*n* = 1), South Africa (*n* = 1), The Netherlands (*n* = 1), United Kingdom (*n* = 4), and United States (*n* = 3). The types of traumatic events included family illness/death (*n* = 6), violence, (*n* = 4), burn injury (*n* = 2), natural disasters (*n* = 2), political detention/deportation (*n* = 2), sexual abuse (*n* = 2), and experiencing cancer (*n* = 1). Thirteen of the 19 studies (68%) had child participants only. The majority of the qualitative data (*n* = 11) were collected using semi-structured interviews (see Table 1).

### 3.3. Main Findings of Qualitative Studies

Traumatised children’s experiences and perspectives towards the traumatic events were categorized in five themes: daily life problems related to trauma, negative responses to trauma, perceived health needs, coping strategies related to trauma and stress, and growth from traumatic experience. These themes with their subthemes are presented in Table 2.

#### 3.3.1. Daily Life Problems Related to Trauma

Traumatic events negatively impacted children’s daily life and relationship issues. Traumatised children experienced disruptions in their daily lives, such as developing poor appetite [15,19,26,27], difficulties with school comprehension [18,27,32], communication problems [30], and disturbances in their daily routines due to frequent medical follow-ups [11,18,26]. Several studies reported that children had found it difficult to maintain interpersonal relationships because they had to move or could not attend most school and social events [19,21,24,25,26]. Moreover, those children experienced vivid flashbacks and intrusive thoughts related to their trauma in the form of sleep disturbances and nightmares, which interfered with their daily functioning [15,16,17,18,20,21,24,25,27]. In injury-related trauma research, injury management measures, such as activity restriction and protective solutions, profoundly impacted the traumatised children’s activities of daily life [24,28]. Many children found injury management measures frustrating and annoying because they stopped them from participating in sports and activities with their friends [24,28]. They sometimes felt that they had lost their independence and needed assistance from caregivers or nurses for self-care tasks such as bathing and toileting [28].

#### 3.3.2. Negative Responses to Trauma

##### Physical Aspect

Most other studies reported traumatised children as being vulnerable to headaches, increased heart rate, stomach ache, trembling, might have had difficulty breathing, dizziness, sleeping difficulties, lethargy, appetite changes, decreased concentration, hyper-alertness, and/or nightmares [11,15,17,18,19,21,26,27,28,29]. One study reported that some children who experienced the loss of a parent felt severe weakness, tightness in their chest, and/or heart palpitations after the traumatic loss [15].

##### Psychological Aspect

There were various negative feelings that were found to be directly related to trauma, such as loss, fright, loneliness, stress, guilt, shame, helplessness, isolation, shock, and being scared, and, fear, anger, worry, and sadness were most frequently mentioned in all included studies. The fear was often related to their loss [25,27,31,32], worrying about the occurrence of the events [18,23,27,28], uncertainty about the future [21], and being in trouble [20]. Most traumatised children reported feeling angry at themselves, others, or at God for some situations such as their traumatic experience and pain, the injustice of their parents’ death or lack of understanding [15,17,19,20,21,24,26,29,30].

Additionally, the most common worries were about the health of their family, threat of violence or the potential outcomes [16,19,21,25,26,27,29]. Those who experienced burn accidents worried about medical procedures because they did not know what to expect and were scared of experiencing pain during the process [18,28]. Traumatised children also experienced persistent sadness, anxiety, and post-traumatic symptoms such as intrusive distressing memories of their injury [16,18,20,21,24,25], and in some cases, refused to express any feelings [15,29].

##### Cognitive Aspect

Traumatic experiences negatively affected children’s cognition, including loss of memory and forgetting aspects of the traumatic accident [20,25]. Children with injury-related experiences tended to ruminate and catastrophise by imagining worst-case scenarios [18,28]. Certain cues could trigger vivid and detailed flashbacks and intrusive thoughts of the trauma [16,17,18,19,20,21,24,25,27,32]. Children with injury-related trauma relived the pain and emotions (e.g., fear) associated with their trauma while experiencing static or moving accident-related visual intrusions [18,24,28]. These intrusions could be triggered by the location of the incident, the specific object that caused the injury, or by watching clips (e.g., fire-related material) on television [18,24,28].

##### Behavioural Aspect

There was a wide variety of changes in children’s behaviours after a traumatic experience such as crying [15,16,17,19,29], increased aggressive or risky behaviours [15,16,17,29], being quiet [17,25,29], insulting others [16,17], loss of motivation [11], being stubborn or angry with siblings and parents [25], taking responsibility for household duties [25], and preferring to stay at home [25]. Traumatised children sometimes showed increased attachment-seeking behaviours that provided comfort, including being close to family or school [15,17,25,29], and seeking their parent’s support [23]. In a study on exposure to domestic violence, the children were found to develop a suicidality when they were unsuccessful in stopping the violent events, leading to a loss of control and powerlessness coupled with the inability to talk about trauma [17].

Additionally, injured children exhibited not only hypervigilant behaviours such as being overly concerned about their surroundings and protective of their scars, but also avoidance behaviours towards situations, objects, and activities that reminded them of their trauma [18,24,28]. Despite receiving approval from their medical team, they preferred to avoid sports and everyday activities, such as riding bicycles, to increase their sense of safety [28]. Conversely, children with other traumatic injuries sometimes found alternative sedentary activities, or devised ways to remain connected to their original hobbies with varying degrees of acceptance [24].

##### Social Aspect

Traumatised children showed negative social reactions. Some bereaved children often avoided all forms of communication with others or withdrew from their social circles [15,29]. Six studies also found that traumatised children experienced social withdrawal due to fear of others’ negative reactions [15,17,19,24,26,29]. Sexually abused children developed negative social reactions because they had become distrustful of others and felt unsafe in their relationships, even with family members and friends [21]. Those who experienced conflict-related trauma reported having difficulty connecting with others and preferred keeping other people at a distance because they felt they were being looked down upon, developed a lack of trust for others or had a fear or expectation of being hurt [17,26]. Furthermore, injured children avoided social interactions because they were worried about what others might think of changes in their physical appearance such as scars, external fixators, or having a limp [19,24]. Their friendships became weaker as they thought their friends no longer understood them and conflicts frequently developed between them [19]. They then tended to establish new friendships with other children with an injury or illness, as they could relate to each other better [19].

#### 3.3.3. Perceived Health Needs

Children who developed post-traumatic stress could have different needs. First, several studies found that the most commonly expressed need among traumatised children was emotional support. They indicated a need to freely talk about their feelings with someone they trusted [30,32]. Families could play a vital role in providing emotional support and physical closeness, especially comforting and reassuring anxious children during medical procedures [11,17,18,24].

Second, social support provided genuine understanding, empathy, and mutual support from peers, health and educational professionals, and communities. Examples included counselling and mentorship at school and church [16]. Two studies showed that brain-injured children liked to communicate with peers undergoing a similar situation and discuss how they handled their problems [25,30]. When support was perceived as pleasant and good, it had a positive effect on the children getting through the traumatic experience.

Third, traumatised children expressed a desire for more detailed information. Because of post-traumatic amnesia or loss of consciousness, children desired having more detailed information than what was suitable to their developmental age [25,30]. Children of parents with brain injury-related experiences expressed their need for more information and advice about living with someone with a brain injury and managing their anxiety [30].

Fourth, traumatised children expressed a desire to play. Children who lived with or had experiences with their injured or ill parents often felt bored, wanted to play with their peers, and returned to their usual activities [24,25]. A study found that play as an activity can dramatically change a traumatised child’s life at home and in hospital [11]. Immediately or sometime after post-injury, children needed emotional support, social connections, and detailed information, which helps them counteract feelings of being abandoned, alone, and anxious [23,25].

#### 3.3.4. Coping Strategies Related to Trauma and Stress

Most of the included studies indicated that traumatised children reported using cognitive and behavioural coping strategies to deal with trauma and stress. They sometimes used more than one coping strategy simultaneously, with each one reinforcing the others. The most used coping strategies are described as follows.

##### Cognitive Coping Strategies

The most commonly reported cognitive coping strategies were positive thinking, distraction, turning to religion and praying, rationalisation, and acceptance. The consistent use of cognitive distancing strategies that involved distraction and cognitive avoidance, such as leisure activities, could help both younger and older children to forget or at least take a break from complicated thoughts about traumatic events, including the loss of a loved one, tsunamis, and parental-illness-related issues [15,16,17,23,25,26,28,29,30]. Making use of positive thinking, mainly focusing on positive outcomes, and using positive affirmations, might help children to calm down and overcome distressing memories [18,23,25,29,30,31]. Moreover, turning to religion and prayer appeared to help children who experienced traumatic events (e.g., violence and natural disasters) obtain safety and comfort by seeking God’s blessing, asking for forgiveness, giving strength, and providing hope for the future [16,23,32]. Rationalisation and acceptance might also help children accept and adapt to their changed life circumstances [23,25,29,30].

##### Behavioural Coping Strategies

The behavioural coping strategies undertaken by children in most studies included behavioural avoidance, talking to others, seeking support, helping others, gradual exposure, suppression of emotions, relaxation, and/or risky and aggressive behaviours. To prevent further distress, traumatised children reported using behavioural avoidance and suppression of emotions such as deliberately not crying, being quiet, and withdrawing socially [15,18,20,22,28,29,30]. In contrast, children with burn-injury-related trauma were more likely to use gradual exposure to face their fears [18]. In a study of people in political detention, early exposure to social activities seemed helpful for children in overcoming their trauma and enhancing feelings of belonging and a sense of commitment to social issues [22]. Talking to others, such as family members and friends, and seeking support were reported to be particularly helpful in dealing with negative feelings [16,17,18,23,26,29,30,32]. Additionally, relaxation techniques, such as deep breathing, were sometimes used to ease participants’ intense emotions [18,30]. Older children who experienced tsunami and parental-illness-related trauma reported that helping people in need made them feel valued [23,25]. On the other hand, children who experienced the sudden loss of a parent and violence-related trauma developed risky behaviours, including stealing, fighting, or other criminal activities, to cope with the pressure and increase their sense of safety [15,16,17,29].

#### 3.3.5. Growth from Traumatic Experience

Traumatic experiences could also produce some positive outcomes and benefits. The experience of trauma could help children to realise the meaning and importance of life, and some were even able to positively reframe and transform their experiences [15,17,24,28,29,31]. Some children reported feeling closer to others, such as their parents [17,25,31]. In addition, trauma could also prompt new life goals and action plans to help others in need and live well by committing to a changed lifestyle [15,18,24,28,31]. Furthermore, children might become more confident, courageous, resilient [15,22,24,31], and more able to overcome difficulties [15,28,31]. Some children became more religious and closer to God by praying and seeking God’s blessing [15,16,23,31].

## 4. Discussion

This systematic review aimed to consolidate the best available current qualitative studies on children’s perspectives of their traumatic experiences and related responses. The findings could help health professionals and researchers integrate the best evidence and most important needs/concerns from these vulnerable groups of children who had traumatic experiences. A total of 19 qualitative studies and their findings were examined and summarised to understand perceived daily-life problems related to trauma, responses to trauma, perceived needs, coping strategies, and growth from trauma experiences. The review consistently found that traumatised children were often living in debilitating and stressful conditions that exacerbated their negative physical, psychological, and behavioural responses to trauma and interfered with their psychological well-being and daily functions, such as with self-care. This finding suggests that paying attention to the traumatised children’s voices and feelings is necessary, alongside the provision of timely care and support.

Parents and peers could play an important role in children’s recovery from traumatic experiences by providing physical closeness and genuine understanding. Children might emphasise the importance of emotional and mutual support from their family and peers after a traumatic event. Children’s distress levels can be reduced when being supported by peers and other social contacts with open sharing, psychological comfort, empathy, and reassurance [33]. Parents are often children’s leading source of support post-trauma, and parental behaviours may mediate children’s resilience [34]. This review elaborated on the traumatised children’s perceptions of family and peer support and encouraged them to engage, participate, and assist in specialised intervention strategies. Furthermore, the findings highlight the need for caregivers and peers to provide further guidance and support during their child’s physical and emotional recovery.

There might be a lack of resources among traumatised children to cope with their situation. The most common coping strategy reported across studies was the persistent avoidance of stimuli that were associated with trauma events. Avoidance coping seemed not to work in the long-term. The negative effect of continued avoidant coping was especially well-examined in research, including a higher level of depression and PTSD symptoms [35,36]. Among the included studies, traumatised children reported a need for support, resources, and information about their situation [25,30]. Therefore, it is important to facilitate traumatised children‘s help-seeking and help them master healthy coping skills. This evidence highlights the need to provide easily accessible trauma-specific services and interventions that help them understand and better cope with their negative psychological responses and related stress towards the traumatic experiences.

Traumatised children often had negative responses and consequences to trauma. However, on the other hand, some could experience positive life changes after trauma, including realising the meaning of life and thinking of survival as a second chance to live better. Consistent with recent studies, growth after trauma could also be found in traumatised children, together with those above-mentioned negative impacts. Mancini et al. [37] found that traumatic experiences could be a reason for survivors to enhance their psychological well-being by self-reflecting and overcoming their difficulties. However, post-traumatic growth, which seems like a coping mechanism and defence against the pathogenic consequences of trauma, lacks adaptive value that reduces distress, depression, and anxiety levels [38]. The extents of their impairments may depend on whether they have had access to mental health services, and/or whether they were kept safe from compounding traumatic events [39]. Given the stress responses in traumatised children who experienced growth, early screening and identification of both positive and negative impacts of traumatic events, with subsequent appropriate and timely intervention designed to strengthen and weaken these impacts respectively, are necessary.

Interviewing children about their responses to traumatic experiences is important for understanding their thoughts and situations. Common subjective experiences and impacts of trauma were revealed. However, getting a child to reflect on their life situation, functioning, and well-being, especially when negative and traumatic in nature, is challenging because children lack verbal and conceptual abilities, and the competence of recall, as well as an overall narrative to share their experiences due to their developmental stages [11,40,41]. When interviewing young children, the presence of parents during the interview may enhance the data as parents can give cues to the child and explain interview questions and scenarios [42]. Interviews with young children should involve caregivers and focus more on behavioural manifestations than on verbal descriptions of their internal state [43]. Furthermore, tailored assessment methods, such as examining verbal and non-verbal expressions, perceptions, and level of understanding, should be considered [40,41]. Future assessments and qualitative research on children and trauma should adopt more tailored and child-friendly methodologies such as pre-meeting interviews with families for building rapport and confidential relationship and individual or parental presence interview depending on the children’s preference [44].

### Strengths and Limitations of the Review

The findings of this systematic review had several implications for practice and future research in the fields of post-traumatic stress and children care. First, the results highlighted the importance of children’s perspectives in assessment and research. As suggested by Coyne [45], conducting research with children was essential and important to gain access to and a clear understanding of the child’s views or make them visible. It highlights the need for early, appropriate, and evidence-based screening, as well as identification and interventions for traumatised children. Second, traumatised children may experience complex events/situations and feelings of loss and stress that need to be shared with their peers to normalise and validate their experiences. To facilitate traumatised children’s sharing of trauma narratives and peer support, practitioners and educators can apply the mutual aid model of group work and trauma-informed care principles in their practice [46]. Third, to provide effective and timely support for traumatised children, caregivers and professionals who are working with traumatised children need to be aware of information such as common post-traumatic stress/PTSD symptoms and guidelines on how to intervene. Fourth, future qualitative research on children and trauma care can compare children’s reactions and coping strategies in diverse samples, such as children with different exposures (single versus multiple; different types/natures and severity), traumatic experiences, and sociocultural backgrounds.

Several methodological limitations of the review need to be considered. First, we only examined qualitative studies in English and might have missed some relevant evidence in other languages. Second, the review did not impose requirements on the minimum levels or varieties of post-traumatic stress symptoms and types of traumatic events, as well as the recall bias induced by a longer period of exposure from the traumatic event. In addition, some studies did not mention the length of time since the children experienced trauma, which might compromise the validity and generalisability of the findings.

## 5. Conclusions

This review has provided comprehensive evidence of the experiences and perspectives of individual traumatised children about their responses, impacts/challenges, needs, and coping with trauma. The review found that traumatic experiences lead to disruptions in traumatised children’s daily lives, mood swings towards traumatic experiences and various aspects such as well-being, mental health, and changes in behaviours. The children appeared to try to manage the psychological stress that arises from trauma through different behavioural and cognitive coping strategies. For instance, they sometimes used persistent avoidance of trauma-related stimuli, which could not work well or was found to be ineffective in long-term. This evidence supports the inclusion of the voice and perspective of children and in both the assessment and intervention regarding post-traumatic stress reaction or disorder. To facilitate traumatised children’s understanding to better cope with their stress and related impacts, there is a pressing need to provide early assessment and appropriate intervention based on these children’s individual concerns and needs to minimise post-traumatic stress symptoms, and thus, the subsequent emergence of PTSD.

## Figures and Tables

**Figure 1 behavsci-13-00170-f001:**
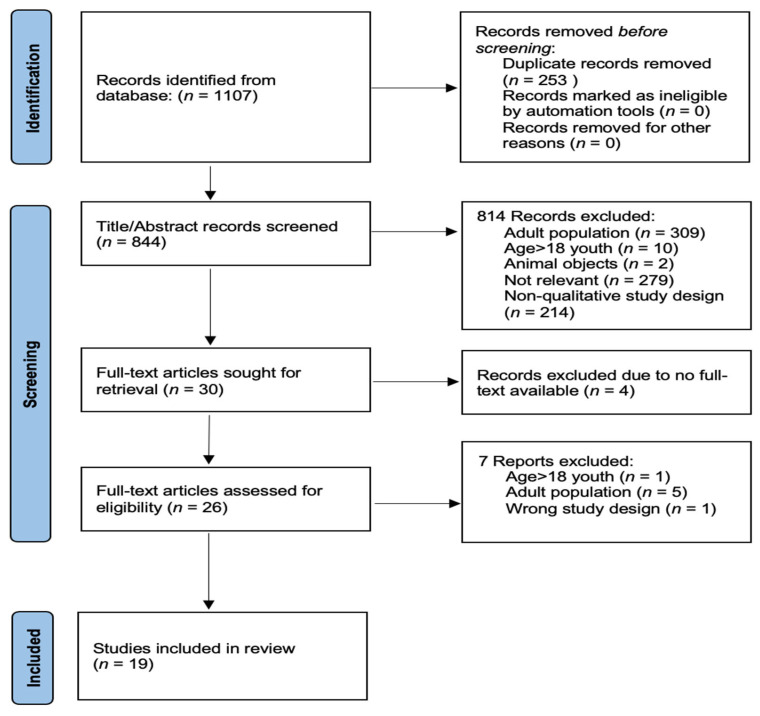
PRISMA flow diagram of the study selection.

**Table 1 behavsci-13-00170-t001:** Summary of included studies.

Study (Authors, Year of Publication and Country)	Type of Trauma	Sample	Study Design	Data Collection Method
Asgari and Naghavi (2020) [15], Iran	Sudden loss of a parent	14 children (age: 14–17)	Qualitative	In-depth and semi-structured interviews
Cherewick et al. (2015) [16], Democratic Republic of Congo	Violence	30 children (age: 10–15)	Qualitative	In-depth interviews (Interview questions given, face-to-face)
Chester and Joscelyne (2021) [17], United Kingdom	Domestic violence	5 children (age: 14–18)	Qualitative	Semi-structured interviews
Darcy et al. (2014) [11], Sweden	Cancer	13 children (age: 1–6) 23 parents	Qualitative	Semi-structured interviews (Face-to-face, parental presence)
Egberts et al. (2020) [18], The Netherlands	Burn injury	8 children (age: 12–17)	Qualitative	Semi-structured interviews (Face-to-face, individual or with parental presence)
Figge et al. (2020) [19], Cambodia	Traumatic experiences (e.g., domestic violence)	30 children (age: 10–13) 30 caregivers	Qualitative	In-person interviews (interview questions given)
Foster (2017) [20], United States	Sexual abuse	19 children (age: 3–17)	Qualitative	Written narratives
Foster and Hagedorn (2014) [21], United States	Sexual abuse	21 children (age: 6–17)	Qualitative	Written narratives
Harazneh et al. (2021) [22], Palestine	Political detention	18 children (age: 12–18)	Qualitative	Semi-structured interviews (Face-to-face, individual or parental presence)
Jensen et al. (2013) [23], Norway	Tsunami	56 children (age: 6–18)	Qualitative	Semi-structured interviews (individual, face-to-face)
Jones et al. (2021) [24], United Kingdom	Traumatic injury	13 children (age: 5–15) 19 parents/guardian	Qualitative	Semi-structured interviews via telephone call or in-person (Joint or separate with parents/guardian)
Kieffer-Kristensen and Johansen (2013) [25], Denmark	Parental ABI	14 children (age: 7–14)	Qualitative	Semi-structured interviews (Face-to-face)
Lee et al. (2018) [26], Myanmar	Traumatic experiences (e.g., conflict and violence)	28 children (age: 12–17) 12 adults (parents/teachers/service providers)	Qualitative	In-depth interviews
Lovato (2019) [27], United States	Forced family separation (parental deportation)	8 children (age: 14–18) 8 mothers 11 school-based staff	Qualitative	Semi-structured interviews (Face-to-face)
McGarry et al. (2014) [28], Australia	Burn injury	12 children (age: 8–15)	Qualitative	In-depth and unstructured interviews (Face-to-face, individual), record all non-verbal cues
Parsons et al. (2021) [29], South Africa	Loss of a parent	22 children (age: 10–12)	Qualitative	Semi-structured interviews (Individual)
Rohleder et al. (2017) [30], United Kingdom	Parental ABI	6 children (age: 9–18) 6 parents (3 with ABI) 3 support workers	Qualitative	Semi-structured interviews (Face-to-face)
Salawali et al. (2020) [31], Indonesia	Natural disasters	16 children (age: 12–18)	Qualitative	In-depth interviews (face-to-face, using field notes)
Tyerman et al. (2019) [32], United Kingdom	The potential loss of the injured sibling with ABI	5 children (age: 9–12)	Qualitative	Semi-structured interviews (Individual or with parental presence)

Note. ABI = acquired brain injury.

**Table 2 behavsci-13-00170-t002:** Themes and subthemes of traumatised children’s experiences and perspectives towards the traumatic events.

Themes	Subthemes
Daily life problems related to trauma	Daily routines [11,15,18,19,24,26,27,28,30]
	Relationship issues [19,21,24,25,26,28,30]
	Daily function [15,16,17,18,20,21,24,25,27,28,32]
Negative responses to trauma	Physical aspect [11,15,17,18,19,21,26,27,28,29]
	Psychological aspect [15,16,17,18,19,20,21,23,24,25,26,27,28,29,30,31,32]
	Cognitive aspect [16,17,18,19,20,21,24,25,27,28,29]
	Behavioural aspect [11,15,16,17,18,19,23,24,25,28,29]
	Social aspect [15,17,19,21,24,26,29]
Perceived health needs	Emotional support [11,17,18,24,30,32]
	Social support [16,25,30]
	More detailed information [25,30]
	Need for play [11,23,24,25]
Coping strategies related to trauma and stress	Cognitive coping strategies [15,16,17,18,23,25,26,28,29,30,31,32]
	Behavioural coping strategies [15,16,17,18,20,22,23,25,26,28,29,30,32]
Growth from traumatic experience	Meaning of life [15,17,24,28,29,31]
	Close relationship [17,25,31]
	New life goals [15,18,24,28,31]
	Personal strengths [15,22,24,28,31]
	Religious beliefs [15,16,23,31]

Note. Examples of illustrative verbatim quotes were shown in Appendix A.

## Data Availability

The data that support the findings of this study are available from the corresponding author upon reasonable request.

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
