# Peer review of "Traumatised Children’s Perspectives on Their Lived Experience: A Review"

_behavsci, 2023, doi:10.3390/bs13020170_

Round 1
Reviewer 1 Report
General comment:
Paper presents a robust work both in its theoretical framework and, above all, its methodological justification. The adopted procedures and sample inclusion and exclusion criteria are clear and well supported. Very relevant and well-structured results. Adjusted discussion that reflects the results.
Sentence corrections required:
1) “Over the past decade, there could be qualitative studies were conducted to explore or attempt to support and guide the 47 development of an appropriate and effective intervention…” (lines 46-47)
2) “…aimed to summarise and consolidate the current evidence from the qualitative evidence research on children’s subjective experiences…” (lines 56-57)
3) “Quality appraisal was not used to determine excluded studies…” (line 96)
4) “One study reported that some children who 173 experienced loss of a parent felt severe weakness, tightness in my chest and/or heart palpitations …” (lines 173-174)
5) “feeling angry at themselves, others, or at God for some situations” (line 184)
6) “understanding. (Asgari & Naghavi, 2020;…) (line 185)
7) “This review elaborated on traumatised children’s perceptions of family and peer support and encouraged them to participate and assist in specialised intervention strategies.” (lines 371-373)
Sugestion:
1) Present table 2 after main findings of qualitative studies – it makes more sense and help to sumarize the authors findings.
Author Response
Thank you for your valuable comments. We have addressed all your comments and suggestions and revised the manuscript accordingly. The details how we have addressed your comments can be found in the attached file.

Reviewer 2 Report
Please see attached file.

Author Response
Thank you for your valuable comments. We have addressed all the comments and made revisions in the manuscript accordingly. The details about how we have addressed your comments are described in the attached file.
